# Morphology Effect of Zinc Oxide Nanoparticles on the Gas Separation Performance of Polyurethane Mixed Matrix Membranes for CO_2_ Recovery from CH_4_, O_2_, and N_2_

**DOI:** 10.3390/membranes12060577

**Published:** 2022-05-31

**Authors:** Tatyana Sergeevna Sazanova, Kirill Alexandrovich Smorodin, Dmitriy Mikhailovich Zarubin, Kseniia Vladimirovna Otvagina, Alexey Andreevich Maslov, Artem Nikolaevich Markov, Diana Georgievna Fukina, Alla Evgenievna Mochalova, Leonid Alexandrovich Mochalov, Artem Anatolevich Atlaskin, Andrey Vladimirovich Vorotyntsev

**Affiliations:** 1Laboratory of Membrane and Catalytic Processes, Nanotechnology and Biotechnology Department, Nizhny Novgorod State Technical University n.a. R.E. Alekseev, Minin Str. 24, 603950 Nizhny Novgorod, Russia; an.vorotyntsev@gmail.com; 2Chemical Engineering Laboratory, Research Institute for Chemistry, Lobachevsky State University of Nizhny Novgorod, Gagarin Ave. 23, 603022 Nizhny Novgorod, Russia; smorodin.kirill.a@gmail.com (K.A.S.); dimazarubin493@gmail.com (D.M.Z.); k.v.otvagina@gmail.com (K.V.O.); the_tardis@inbox.ru (A.A.M.); markov.art.nik@gmail.com (A.N.M.); dianafuk@yandex.ru (D.G.F.); mochalova_ae@mail.ru (A.E.M.); mochalovleo@gmail.com (L.A.M.); 3Laboratory of SMART Polymeric Materials and Technologies, Mendeleev University of Chemical Technology, Miusskaya Sq. 9, 125047 Moscow, Russia; atlaskin.a.a@muctr.ru

**Keywords:** membranes, gas separation, polyurethane, zinc oxide, nanoparticles

## Abstract

The effect of the morphology and content of zinc oxide nanoparticles (ZnO-NPs) on the physicochemical, mechanical, and gas transport properties of the polyurethane (PU) mixed matrix membranes (MMMs) with respect to CO_2_ recovery from CH_4_, O_2_, and N_2_ was studied. The MMMs based on PU with spherical and rod-shaped ZnO-NPs at various loadings, namely, 0.05, 0.1, 0.5, 1, and 2 wt. %, were prepared with membrane density control and studied using AFM, wettability measurements, surface free energy calculation, gas separation and mechanical testing. To evaluate the resistance of the ZnO-NPs to agglomeration in the polymer solutions, zeta potential was determined. The ZnO-NPs with average cross sectional size of 30 nm were obtained by plasma-enhanced chemical vapor deposition (PECVD) from elemental high-purity zinc in a zinc-oxygen-hydrogen plasma-forming gas mixture. It was established that the spherical ZnO-NPs are promising to improve the gas performance of PU-based MMMs for CO_2_ recovery from natural gas, while the rod-shaped NPs better demonstrate their potential in capturing CO_2_ in flue gases.

## 1. Introduction

Nowadays, one of the most promising and energy efficient approaches to carbon dioxide recovery from natural [1,2,3] and flue [3,4,5,6] gases is membrane gas separation providing the process in the absence of phase transitions at room temperature.

To implement the membrane gas separation technology, inorganic and polymeric materials are used. As a rule, inorganic membranes have exceptional chemical, mechanical and thermal stability, exhibit higher gas flows and can withstand higher pressures in contrast to polymeric ones [7]. However, their higher cost and ability to be processed into modules for large-scale applications are the main challenges in using these materials for gas separation [7,8]. In this context, polymeric membranes are the mainstay of commercialization due to their ease of processing, manufacturing, and availability. However, the use of polymeric membranes does not always solve the problem of obtaining the optimal ratio of selectivity and permeability. So, researchers resort to using various composite and hybrid membranes [9,10,11,12,13]. One such approach is the use of mixed matrix membranes (MMMs) by loading an inorganic (usually nanosized) filler into a polymer matrix [14,15,16,17,18,19,20]. This way is the most versatile and relatively simple to implement due to the possibility of improving the gas separation properties (as a rule, selectivity) of existing polymeric membranes, as well as allows to amalgamate the benefits of both polymeric and inorganic materials.

MMMs can be filled with a variety of inorganic materials, such as metal oxides [20,21,22,23,24,25], metal organic frameworks (MOFs) [26,27,28,29,30], carbon molecular sieves (CMS) [31,32,33,34,35], carbon nanotubes (CNTs) [36,37,38,39,40], and zeolites [15,41,42,43]. Nevertheless, regardless of a filler type, the manufacture of MMMs is often associated with a number of difficulties, such as agglomeration of inorganic fillers, their sedimentation and insufficient dispersion [7]. The dispersion of inorganic fillers in a polymer can be improved by applying the priming method [44,45,46], interfacial polymerization processes [47,48,49], as well as additional agents to increase adhesion at the polymer/filler phase interface [50,51,52]. Moreover, the overall preparation of MMMs is also influenced by the size and morphology of the filler particles, the rigidity of the polymer chain, as well as the presence of affinity and interaction between the introduced phase and removed gas. Thus, the key to the successful preparation of MMMs is the selection of the suitable polymer/filler pair [53,54,55].

Based on the acidic nature of carbon dioxide, which can act as an electron acceptor (Lewis acid), the introduction of an electron donor (Lewis base) into the polymer matrix can increase the selectivity of the membrane without reducing its permeability. In this term, promising filler for polymeric membranes is zinc oxide nanoparticles, which form a Lewis pair with carbon dioxide. In addition, zinc oxide is insoluble in water. This fact prevents the dissolution of the filler upon contact with wet gases.

The effectiveness of zinc oxide as an MMMs nanofiller for CO_2_ recovery has already been proven in a number of works [24,56,57,58]. So, Farashi Z. and co-workers [24] showed that the CO_2_ permeability and ideal CO_2_/CH_4_ selectivity increased about 13% and 21%, respectively, at loading 10 wt. % of zinc oxide nanoparticles into the polymer matrix based on polyether-block-amide (Pebax-1657). The spherical zinc oxide nanoparticles had an average particle size of 18 nm. The membrane performance was evaluated at a pressure of 3 bars and temperature of 30 °C. In another work, Azizi N. and co-workers [58] studied effect of zinc oxide on the performance of membranes based on poly (ether-block-amide) (Pebax-1074). The authors showed that the fabricated nanocomposite membranes exhibited better separation performance compared to the neat Pebax-1074 membrane in terms of both permeability and selectivity. As an example, CO_2_ permeability, ideal CO_2_/CH_4_ and CO_2_/N_2_ selectivity values for the neat Pebax-1074 membrane were 110.67 Barrer, 11.09 and 50.08, respectively, while those values were 152.27 Barrer, 13.52 and 62.15 for the nanocomposite membrane containing 8 wt. % of zinc oxide nanoparticles. The spherical zinc oxide nanoparticles had an average particle size of 10-30 nm. The membrane performance was evaluated at a pressure of 3 bars and temperature of 25 °C.

However, it is important to note that the filler effectiveness depends not only on the content and size of nanoparticles, but also on their purity and morphology. This fact is associated with a number of reasons. First, the presence of any impurities in the filler introduced into the polymeric matrix can lead to its uncontrolled interaction not only with the removed carbon dioxide but also with methane, nitrogen, or oxygen and to a decrease in the efficiency of membrane separation. Secondly, the nanoscale of zinc oxide affects the behavior of electrons in the resulting material [59], which is important for its interacts with carbon dioxide, since, this can affect membrane selective properties. Thirdly, nanoparticles loaded into a polymer matrix can lead to uncontrolled changes in the fractional free volume of the membrane depending on the mobility and rigidity of macromolecular chains and, accordingly, its permeability for each component in a gas mixture. Degree of those changes depends on both the size and morphology of the loaded nanophase.

For the first time in membrane science, this study represents the effect of the morphology of zinc oxide nanoparticles (ZnO-NPs) and their content in a polymeric matrix on the physicochemical, mechanical, and gas transport properties of polyurethane (PU) MMMs with respect to CO_2_ recovery from CH_4_, O_2_, and N_2_.

## 2. Experimental Section

### 2.1. Materials

PU Elastollan^®^ 1190 A were purchased from BASF (Ludwigshafen, Germany). Tetrahydrofuran (THF) were supplied from Chimreactive (Nizhny Novgorod, Russia) and purified by distillation before using. Liquids for wettability tests and density measurements (diiodomethane (99%), glycerol (99%), and ethylene glycol (99.5%)) were purchased from Merck (Darmstadt, Germany).

High purity gases (carbon dioxide (99.99%), methane (99.99%), nitrogen (99.999%), and oxygen (99.7%)) for gas separation testing were supplied from Horst Technologies Ltd. (Dzerzhinsk, Nizhny Novgorod region, Russia). 

Zinc of 5N purity (Changsha Rich Nonferrous Metals Co., Ltd., Hunan, Changsha, China), high purity hydrogen (99.9999%) and high purity oxygen 6.0 (99.9999%) (Horst Technologies Ltd., Dzerzhinsk, Nizhny Novgorod region, Russia) were used as components of a plasma-forming mixture. Hydrogen was used as a carrier gas, which also acted as a temperature stabilizer and regulator of the growth of nanostructures.

### 2.2. Plasma-Chemical Synthesis of ZnO-NPs

Spherical and rod-shaped ZnO-NPs were obtained by plasma-enhanced chemical vapor deposition (PECVD) under optical emission spectrometry control from elemental high-purity zinc in a zinc-oxygen-hydrogen plasma-forming gas mixture. The installation for the plasma-chemical synthesis was described in [60,61]. 

As the first step, the installation was evacuated to a pressure of 1 × 10^−3^ Pa for several hours to remove traces of nitrogen and water from the walls of the reactor. After that, the source with zinc was heated to 370 °C in the case of the spherical NPs and to 470 °C in the case of the rod-shaped ones. The temperature of the collecting powder tank was maintained at a level of 250 °C. Until the system entered the operating mode, the powder tank was closed with a magnetic diaphragm of a special design. The total gas flow through the plasma-chemical reactor was set equal to 30 mL min^−1^ at a total pressure in the system of 0.1 Pa. The plasma discharge power was 100 W in the case of the spherical NPs and 70 W in the case of the rod-shaped ones.

### 2.3. Preparation of Membranes

PU membranes without and with loading ZnO-NPs were fabricated by corresponding polymer solutions casting using an automatic casting machine MemcastPlus (POROMETER, Nazareth, Belgium) onto a glass substrate under ambient conditions (20 °C, 10^5^ Pa). The polymer membranes were easily peeled off the substrates after solvent evaporation and desiccated under vacuum for reaching the constant mass. The thickness of the obtained membranes was about 60 μm.

THF volume of required to obtain 5 wt. % PU solutions was divided into two parts. In the first one, the polymer was dissolved at 70 °C with the help of a heated stirrer; in the second one, ZnO-NPs were dispersed in an ultrasonic bath for 10 min. After that, both parts of THF were mixed with stirring and further ultrasonication for 10 min. The content of ZnO-NPs (0.05, 0.1, 0.5, 1, and 2 wt. %) was calculated relative to the weight of PU.

As a result, the following series of samples was obtained: PU (without adding ZnO-NPs), PU/ZnO-NPs(sph) in the case of the spherical NPs, and PU/ZnO-NPs(rod) in the case of the rod-shaped ones with ZnO-NPs loadings of 0.05, 0.1, 0.5, 1, and 2 wt. %.

### 2.4. Zeta Potential

The resistance of the ZnO-NPs to agglomeration in the polymer solutions was evaluated by determining zeta potential. The measurements were conducted with a generation zeta potential instrument Stabino^®^II (Microtrac Inc., Montgomeryville, PA, USA) using a streaming potential method. The prepared solution of PU/ZnO-NPs in THF was additionally ultrasonicated for dispersion and degassing. The samples were measured for 10 min with a step of 10 s; the obtained values were averaged for each time interval.

### 2.5. Density Measurements

To control the densities of the membranes, the “flotation” method was used. The measurements were conducted as follows: a graduated cylinder (25 cm^3^) with ground joint and the plug was filed to a half with a mixture of ethylene glycol (ρ = 1.11 g/cm^3^ @ 20 °C) and glycerol (ρ = 1.26 g/cm^3^ @ 20 °C) in the volume ratio 1:1. These liquids were chosen because they do not react with the measured samples and do not cause swelling and solvation [62]. After that, a piece of a membrane sample was put down into the cylinder. Based on the position of the membrane sample in the liquid, more of a low density or a high-density liquid was added and mixed with the glass mixer so that a membrane sample takes an equilibrium position in the middle of the liquid column according to the graduation. Then a density of the obtained mixture of liquids was measured using a pycnometer (10 cm^3^). Density was measured five times on each membrane.

### 2.6. Atomic Force Microscopy

The surface of the membranes was studied by atomic force microscopy (AFM) using a scanning probe microscope SPM-9700 (Shimadzu, Kyoto, Japan). AFM scanning was performed using a tapping mode by silicon vibrating cantilevers PointProbe FMR-20 (NanoWorld Innovative Technologies, Neuchâtel, Switzerland) with a stiffness coefficient of 1.3 N/m and a typical tip radius of no more than 8 nm (guaranteed—no more than 12 nm); a tip height was 15 μm. After AFM scanning, an arithmetic average roughness height (R_a_) and a mean roughness depth (R_z_) were obtained. A base length was 10 μm. Processing of the obtained AFM images and their analysis were performed using a software SPM Manager ver. 4.02 (Shimadzu, Kyoto, Japan).

### 2.7. Scanning Electron Microscopy and Energy Dispersive X-ray Spectroscopy

The stoichiometry and size-morphological characteristics of the ZnO-NPs were studied by scanning electron microscopy (SEM) and energy dispersive X-ray spectroscopy (EDS) using an electron microscope JSM-IT300LV (JEOL, Peabody, MA, USA) with an electron probe diameter of about 5 nm and a probe current of less than 0.5 nA (operating voltage 20 kV). SEM scanning was performed using low-energy secondary electrons and backscattered electrons under a low vacuum to eliminate the charge.

### 2.8. Wettability Measurements and Surface Free Energy Calculation

Wettability tests were carried out with using three test liquids with different surface tensions (Table 1): water, glycerol, and diiodomethane. A drop of test liquid was applied on a membrane surface in an enclosed area filled with the corresponding liquid’s vapor. After reaching the equilibrium state, a drop image was taken, and the contact angle of wetting (θ) was determined using ImageJ software with contact angle plugin. The measurements were conducted at 20 °C. The results were collected for a series of five drops with contact angle deviations did not exceed ±1°.

Total surface free energy and its polar and dispersive components were calculated based on wettability test results using the Owens-Wendt method [63,64]. 

### 2.9. Gas Separation Experiments

The pure gas permeability coefficients (*P*) of CO_2_, CH_4_, O_2_, and N_2_ through the membranes were measured by an experimental setup (Figure 1) equipped with an automatic computing system based on a software-logic controller V130-33-RA22 (Unitronics, Airport City, Israel) at the initial transmembrane pressure of 110 kPa and ambient temperature (20 °C) in a constant volume mode. Each single-gas test was repeated at least three times. The selectivity (α) of the polymeric membranes was calculated as the ratio of the single gas permeability coefficients.

### 2.10. Mechanical Testing

The mechanical properties, namely, breaking strength (σ, MPa) and elongation at break (ε, %), were measured using Universal Tensile Machine Z005 (ZWICK, Ennepetal, Germany) in tension elongation experiments at a tension rate of 50 mm/min under ambient conditions. The polymer samples were prepared as a flat rectangular shape (50 mm × 10 mm) with original film thickness. The data were collected for ten measurements for each membrane sample and processed with ZWICK Tensile Machine software testControl II (ZWICK, Ennepetal, Germany).

## 3. Results and Discussion

### 3.1. Morphology of ZnO-NPs

Figure 2 shows the SEM images of the spherical and rod-shaped ZnO-NPs obtained by the PECVD method; the average size of the NPs in cross section 30 nm with coefficients of variation of 17% and 23%, respectively.

The EDS analysis data for the obtained ZnO-NPs are shown in Table 2.

### 3.2. Zeta Potential of Polymer Solutions

According to the zeta potential measurements (Figure 3), the negative charge in the polymer solutions was declined due to loading ZnO-NPs. Moreover, the charge drop was more noticeable for the rod-shaped NPs in contrast to the spherical ones. However, in all cases, the value of the zeta potential did not decrease below 25 mV modulo, so, it could be concluded that the ZnO-NPs are well resistant to agglomeration in the polymer solutions and the latter have a high degree of stability [65].

### 3.3. Membrane Surface

The AFM data of the membranes based on the various PU solutions with ZnO-NPs are shown in Figure 4.

The surface of the samples is loosely packed but with low roughness. It is noteworthy that, first, the surface irregularities decreased with loading ZnO-NPs of both types and, then, increased with the growth in the NPs content above 0.5 wt. %. This effect could be related to the compensation of voids in the polymer due to the NPs. Moreover, the smoothing of the membrane surface was more expressed in the case of loading the spherical ZnO-NPs in contrast to the rod-shaped ones. This difference might be due to the varying degrees of resistance to agglomeration of the NPs with different shapes in the initial polymer solutions (Figure 3). The higher the stability of the system, the lower the degree of the NPs agglomeration and the smaller the perturbations introduced into the surface relief formation of the membranes.

It should be noted that the results of the AFM study for the reverse side of the polymer membranes were similar (Appendix A), which indicated the absence or minimal presence of the NPs sedimentation process during film formation.

According to the wettability measurements, it was found that loading the ZnO-NPs into the polymer matrix caused not only a change in the relief of the membrane surface but also a redistribution of polar and nonpolar fragments on it. This led to a change in the nature of the surface wettability of the PU membranes (Table 3).

It is noteworthy that with an increase in the NPs’ loading into the polymer matrix, the wettability of the membranes’ surface with the polar liquids (water, glycerol) mainly decreased and then increased again. The opposite tendency was observed in the case of nonpolar diiodomethane. Such behavior could be associated with a combination of two possible factors, namely, the ZnO presence on the surface of the membranes and a change in their roughness. Probably, the first factor contributed due to the self-energy of the ZnO surface, and the second contributed due to the redistribution of the PU polar and non-polar segments during a physical change in the relief of the membrane.

Based on the wettability data, the specific surface free energy (with its polar and dispersive components) of the polymeric membranes was calculated using the Owens-Wendt method (Figure 5).

With loading spherical ZnO-NPs into the PU matrix, a decrease in the polar surface energy component was observed with a minimum at 0.5 wt. % of the NPs. The same minimum was observed with loading the rod-shaped ZnO-NPs. However, the maximum values slightly exceeded the values of the polar energy for the pure PU membrane.

At the same time, the dispersion surface energy component in the case of both NPs’ types decreased with respect to the pure PU membrane. Such tendency might be due to an increase in the fractional free volume (FFV) [66] of the polymer membranes with loading the ZnO-NPs. This fact is indirectly confirmed by the theory described in the works [63,67] and a decrease in membrane density after modification (Table 4). 

It should be noted that the membrane densities’ decrease with loading the ZnO NPs into the PU matrix was probably associated with an intensification of the NPs’ agglomeration from an increase in their content. With an increase of the NPs’ agglomeration, a forced rearrangement of PU macromolecules probably occurred, which caused a decrease in the density of the membranes.

The essence of the theory mentioned above lies in the inversely dependence between the dispersion component of the surface free energy in polymeric membranes and their FFV, which has the directly dependence with the total gas permeability [63,67]. The fact is that the dispersion component is understood as the sum of van der Waals and other non-site-specific interactions between a polymer surface and an applied liquid. I.e., the dispersion component has only physical origins and depends on the morphological/supramolecular structure of the polymer. The looser the packed structure, the smaller the dispersion component of its surface energy. At the same time, the polar component of the surface free energy in polymeric membranes is understood as the sum of site-specific interactions such as electrostatic, hydrogen bonding, etc., between a polymer surface and an applied liquid. I.e., a change in the polar component can be associated with the corresponding redistribution of polar/nonpolar segments of polymer chains in membranes. Thus, an increase in the polar surface energy indicates an increase in the polar segment proportion on the polymer surface, and vice versa. In this case, the presence of polar segments of the polymer chain on the membrane surface will directly affect its interaction with separated gases. At that, plasticizing gases are more susceptible to this effect than non-plasticizing ones, since they are prone to site-specific interactions.

### 3.4. Gas Separation Properties of Membranes

As exhibited in Figure 6, at all membrane samples, the gas permeability of CO_2_ was higher than those of CH_4_, O_2_, and N_2_. Such tendency was probably associated with intrinsic affinity of ZnO which could render its NPs to interact with CO_2_ more than the other three gases and consequently led higher adsorption capacity for CO_2_ [58,68].

In addition to the affinity of the gases to the membrane filler, the difference in the kinetic diameters of the gas molecules could also made a contribution (Table 5).

However, the different degrees of the changes in the permeability with the ZnO NPs loading into the PU matrix indicated that the affinity of the gases to the membrane filler had the greatest contribution to the formation of the gas permeability.

Notably, loading the ZnO-NPs into the PU matrix caused an increase in the CO_2_ permeability of the polymer membranes with a further decrease below the permeability of pure PU (Figure 6). The maximum growth occurred at 0.5 wt. % NPs loading. Moreover, the peaks of the CO_2_ permeability for the membranes were 30% and 38% in the case of the spherical and rod-shaped ZnO-NPs, respectively. 

The maximum increase in the permeability of other gases was also observed at 0.5 wt. % NPs loading into the polymer matrix. However, the peaks in the case of these gases were relatively modest (Figure 6).

The gas permeability data correlate well with the changes in the specific surface free energy of the membranes (Figure 5) in the context of the theory described above. At that, the maximum permeability of the gases coincided with the minimum of the polar component for the surface energy of the membranes. I.e., a lower value of the polar energy provided less resistance to the interaction between the gases and composite media. As noted earlier, the degree of such influence is higher for the gases prone to site-specific interactions, such as CO_2_ and CH_4_.

However, the changes in the membrane densities should also be taken into account (Table 4). The density of the membranes with the loading rod-like ZnO NPs into the PU matrix exhibited greater degradation than in the case of the spherical ones. This fact could been a reason of the higher CO_2_ and CH_4_ permeability values observed with the loading rod-like ZnO NPs. The opposite tendency for the O_2_ and N_2_ permeabilities could been observed due to the larger molecule kinetic diameter of these gases (Table 5) compared to CO_2_ and the lower propensity to site-specific interactions compared to CH_4_.

As a result of the different degrees of the changes in the permeability with the ZnO NPs loading into the PU matrix, the membranes’ selectivity increased after modification (Figure 7). For both forms of the NPs, the growth peak occurred at 0.5 wt. % NPs loading. It is noteworthy that, the largest increase was observed for CO_2_/CH_4_ in the case of the spherical ZnO-NPs and amounted to 8%. In the case of the rod-like ZnO-NPs, the CO_2_/CH_4_ selectivity remained virtually unchanged. However, the CO_2_/N_2_ and CO_2_/O_2_ selectivities of the PU/ZnO-NPs(rod) membranes increased by 36%, as well as ones increased by 10% and 19%, respectively, in the case of the spherical NPs. 

### 3.5. Mechanical Properties of Membranes

Mechanical tests showed that loading the ZnO-NPs into the PU matrix improved tensile strength and elongation at break of the MMMs with a further decrease in these values (Figure 8).

The greatest degradation of the mechanical characteristics for the polymeric membranes was observed with loading the rod-shaped ZnO-NPs. Probably, that was due to the lower resistance of these NPs to agglomeration (Figure 3) and their elongated shape, which might prevent compact embedding the NPs into the PU matrix. Nevertheless, at an optimal (according to the gas transport tests) ZnO-NPs loading of 0.5 wt. %, the tensile strength of the polymer membranes exceeded one for pure PU.

## 4. Conclusions

In this work, the effect of the morphology of the ZnO-NPs and their content in the polymeric matrix on the physicochemical, mechanical, and gas transport properties of the PU MMMs with respect to CO_2_ recovery from CH_4_, O_2_, and N_2_ was studied.

The MMMs based on PU and the spherical and rod-shaped ZnO-NPs with various NPs loadings, namely, 0.05, 0.1, 0.5, 1, and 2 wt. % relative to the weight of PU, were prepared with membrane density control and studied using AFM, wettability measurements, surface free energy calculation, gas separation and mechanical testing. To evaluate the resistance of the ZnO-NPs to agglomeration in the polymer solutions, zeta potential was determined. Each sample demonstrated a high degree of stability.

It was shown that loading the ZnO-NPs into the PU matrix improved the CO_2_ permeability of the polymer membranes with a further decrease below the permeability of pure PU. The maximum growth corresponded to 0.5 wt. % NPs loading. The peaks of the CO_2_ permeability for the MMMs were 30% and 38% in the case of the spherical and rod-shaped ZnO-NPs, respectively. The permeability peaks of other gases were also observed at 0.5 wt. % NPs loading into the polymer matrix but were relatively modest. The selectivity of the membranes increased too with loading the ZnO-NPs into the polymer matrix. The growth peaks corresponded to 0.5 wt. % NPs loading. It is noteworthy that, the largest increase was observed for CO_2_/CH_4_ in the case of the spherical ZnO-NPs and amounted to 8%. In the case of the rod-like ZnO-NPs, the CO_2_/CH_4_ selectivity remained virtually unchanged. However, the CO_2_/O_2_ and CO_2_/N_2_ selectivities of the PU/ZnO-NPs(rod) membranes increased by 36%, as well as ones increased by 19% and 10%, respectively, in the case of the spherical NPs. 

According to mechanical testing, it was shown that loading the ZnO-NPs to the PU matrix improved tensile strength and elongation at break of the MMMs with a further decrease in these values. The greatest degradation of the mechanical properties for the MMMs was observed in the case of loading the rod-shaped ZnO-NPs. Nevertheless, at an optimal (based on the gas separation data) 0.5 wt. % NPs loading, the tensile strength of the polymer membranes exceeded one for pure PU.

To sum up, it was established that use of spherical ZnO-NPs is promising for the improvement of the gas performance of PU-based MMMs for CO_2_ recovery from natural gas, while the rod-shaped NPs better demonstrate their potential in capturing CO_2_ in flue gases.

## Figures and Tables

**Figure 1 membranes-12-00577-f001:**
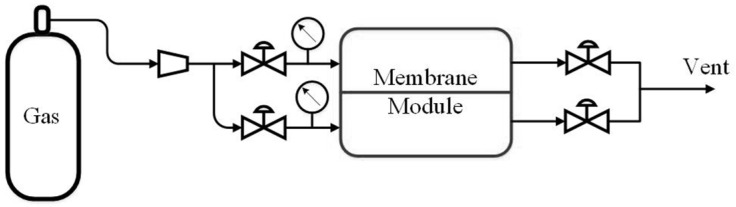
The scheme of the experimental setup for gas separation testing.

**Figure 2 membranes-12-00577-f002:**
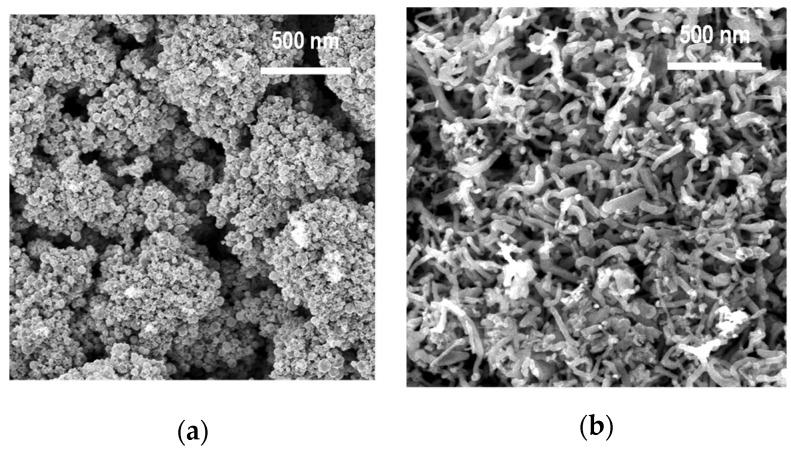
The SEM images of the spherical (a) and rod-shaped (b) ZnO-NPs obtained by the PECVD method.

**Figure 3 membranes-12-00577-f003:**
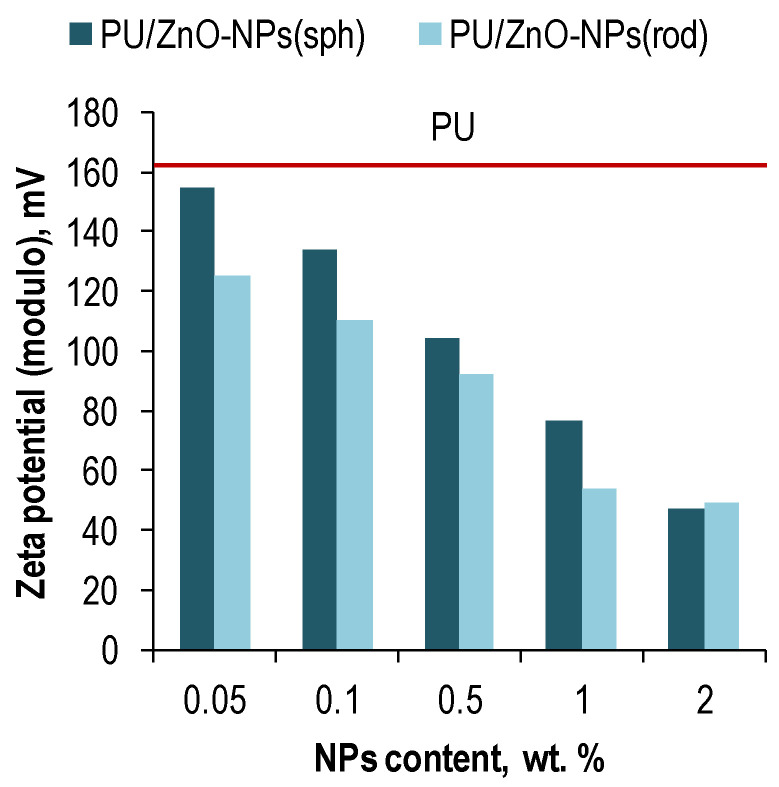
The zeta potential of the polymer solutions based on PU/ZnO-NPs in THF.

**Figure 4 membranes-12-00577-f004:**
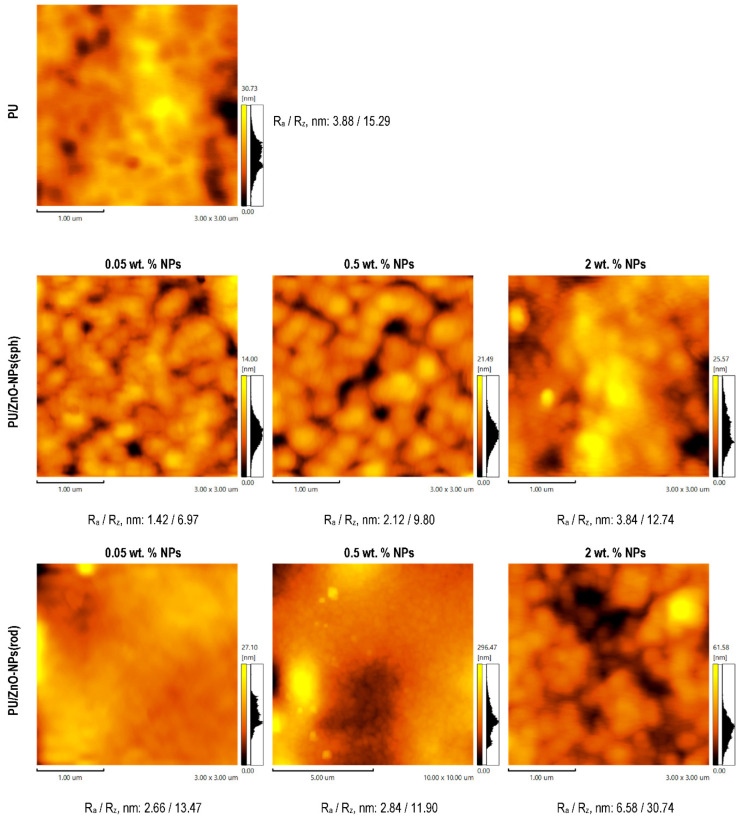
The AFM data of the membranes based on the various PU solutions with ZnO-NPs.

**Figure 5 membranes-12-00577-f005:**
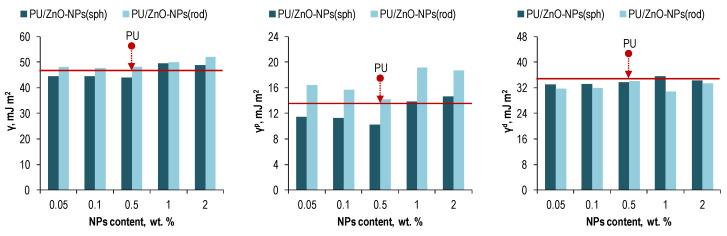
The specific surface free energy (overall (**left**), polar (**center**), and dispersive (**right**)).

**Figure 6 membranes-12-00577-f006:**
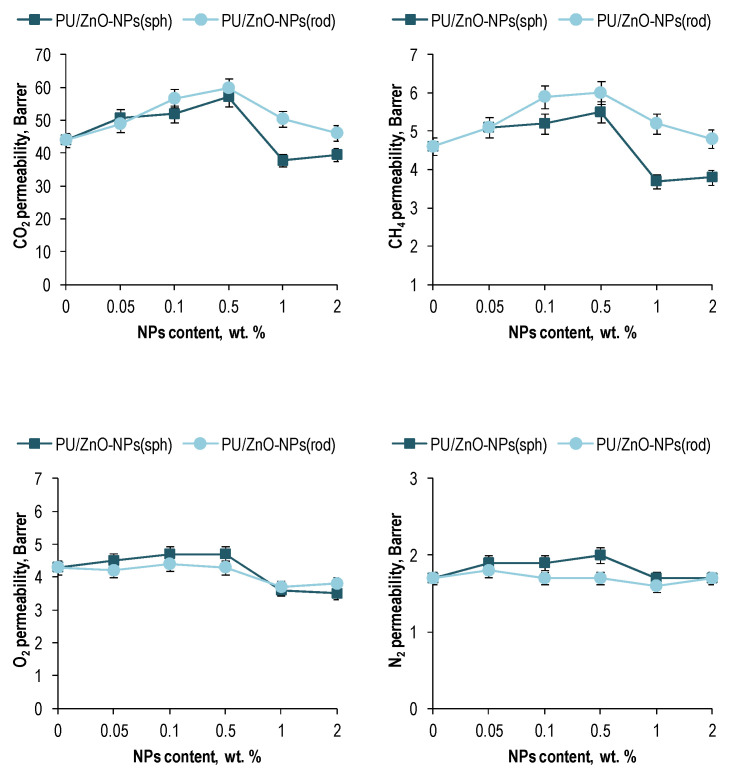
The permeability coefficients for the polymeric membranes based on the PU/ZnO-NPs.

**Figure 7 membranes-12-00577-f007:**
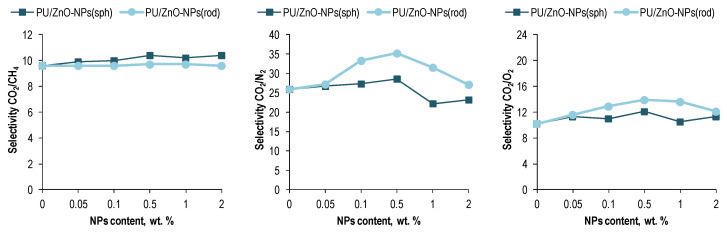
The selectivity of the polymeric membranes based on the PU/ZnO-NPs.

**Figure 8 membranes-12-00577-f008:**
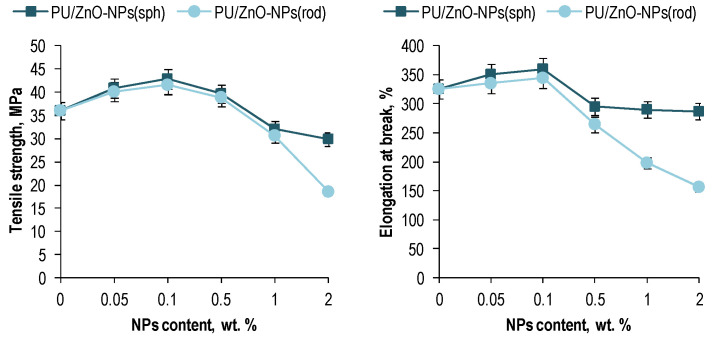
The mechanical properties of the polymeric membranes based on the PU/ZnO-NPs.

**Table 1 membranes-12-00577-t001:** The surface tension values (overall, dispersive and polar) of the test liquids.

Test Liquid	$\gamma_{l},\;\mathrm{mJ}\cdot m^{-2}$	$\gamma_{l}^{d},\;\mathrm{mJ}\cdot m^{-2}$	$\gamma_{l}^{p},\;\mathrm{mJ}\cdot m^{-2}$
Water	72.1	19.9	52.2
Glycerol	63.4	37.0	26.4
Diiodomethane	50.8	49.5	1.3

**Table 2 membranes-12-00577-t002:** The EDS analysis data for the spherical and rod-shaped ZnO-NPs.

Sample	Zn, at. %	O, at. %
ZnO-NPs(sph)	51.2	48.8
ZnO-NPs(rod)	55.7	44.3

**Table 3 membranes-12-00577-t003:** The contact angles of the polymeric membranes.

Sample	NPs Content, wt. %	Contact Angles θ,±1°		
		Water	Glycerol	Diiodomethane
PU	0	62	52	30
PU/ZnO-NPs(sph)	0.05	62	67	19
	0.1	63	65	23
	0.5	68	54	38
	1	56	70	32
	2	58	52	27
PU/ZnO-NPs(rod)	0.05	54	62	19
	0.1	55	63	18
	0.5	58	52	27
	1	50	59	21
	2	50	53	18

**Table 4 membranes-12-00577-t004:** The density of the polymeric membranes.

Sample	NPs Content, wt. %	ρ, g/cm^3^
PU	0	1.15 ± 0.02
PU/ZnO-NPs(sph)	0.05	1.14 ± 0.01
	0.1	1.14 ± 0.02
	0.5	1.14 ± 0.02
	1	1.13 ± 0.01
	2	1.10 ± 0.03
PU/ZnO-NPs(rod)	0.05	1.14 ± 0.02
	0.1	1.14 ± 0.03
	0.5	1.12 ± 0.02
	1	1.11 ± 0.01
	2	1.11 ± 0.02

**Table 5 membranes-12-00577-t005:** Kinetic diameters for different gases [69].

Gases	Kinetic Diameter, Å
CO_2_	3.30
CH_4_	3.80
O_2_	3.46
N_2_	3.64

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
