# Peer review of "Morphology Effect of Zinc Oxide Nanoparticles on the Gas Separation Performance of Polyurethane Mixed Matrix Membranes for CO2 Recovery from CH4, O2, and N2"

_membranes, 2022, doi:10.3390/membranes12060577_

Round 1

Reviewer 1 Report

The article by Sazanova et al. reports a mixed matrix membrane formulation of ZnO nanoparticles with Polyurethane primarily for CO2 separation applications. The use of mixed matrix membranes is significant for current membrane and adsorbent scientists, which makes the topic appropriate for Membranes. Unfortunately, the article has numerous scientific and writing deficiencies that render it unpublishable in its submitted state. The authors highlight that the novelty of their work is the analysis of nanoparticle morphology effects. Before expanding upon specific details, it is worth commenting, that without proper statistical analysis and corresponding detailed discussion, concrete conclusions about nanoparticle morphology have not been reached. Despite my recommendation of rejecting the article, I would like to convey that developing a detailed understanding of NP morphology in MMMs is an important topic. I expand on a number of relevant points below in bulleted list format.

  • A major issue of the submitted manuscript is regarding the text presented in the Introduction. As previously alluded to, the field of mixed matrix membranes (MMM) is quickly evolving. The authors make little connection with other MMM work; such supporting text is key for placing the presented results in context of current separation membrane studies. This point is especially important when considering the popularity of ZnO nanoparticle fillers in MMMs.
  • Continuing with the introduction, there are several statements that are factually inaccurate or too general. As an example, the authors’ state “To implement the membrane gas separation technology, due to availability and necessary properties, as a rule, polymeric materials are used.” This statement neglects the use of carbon molecular sieves, zeolitic membrane materials, and so on.
  • I believe that the results and analysis reported in this work should be expanded. As it stands, several plots are presented but connection with the main motivation and novelty of the work is absent. The discussion surrounding observations should be substantially expanded.
  • Some of the authors’ presented discussion is speculatory. For example, “Such tendency might be due to an increase in the fractional free volume (FFV) [28] of the polymer membranes upon adding the ZnO-NPs. This fact is indirectly confirmed by the theory described in the works [25,29] and a decrease in membrane density after modification (Tab. 4).” If this statement was confirmable within statistical significance, then there would be appreciable difference in the observed permeability. Considering the logarithmic scale of Figure 5, this does not appear to be the case.
  • Sticking with Figure 5, there should be some discussion of the uncertainty associated with these datapoints. Without error bars on these plots, or any mention of their magnitude, it is not possible to judge if the impact of NP content and shape yields a statistically significant difference in membrane permeability.
  • For table 3, I am curious about the determination of +- 1 degree for the contact angle uncertainty. Is this an estimate? A discussion about this (in the methodology) is appropriate. It is also not clear if this is 95% confidence interval or the measurement standard deviation.
  • Figure 7 is lacking uncertainty with each datapoint. Tensile testing should be performed in replicate and results reported as an average with uncertainty. The collection of accompanying stress-strain curves should be supplied in the supplementary information. In my opinion elongation at break and tensile strength at break are somewhat strange mechanical properties to demonstrate membrane durability. I am not aware of any operational conditions where a membrane will experience 150-300% elongation. Perhaps the stress-strain curves are more appropriate for a main text figure.

Author Response

Dear Reviewer,

We would like to thank you very much for the attention paid to our manuscript and priceless scientific comments. We appreciate a significant work done on this manuscript with detailed analysis that was very valuable in order to change the text and improve the manuscript. Your comments were attentively examined and answers were compiled with the relevant amendments to the text of the manuscript.

The responds to the comments are as follows:

  1. A major issue of the submitted manuscript is regarding the text presented in the Introduction. As previously alluded to, the field of mixed matrix membranes (MMM) is quickly evolving. The authors make little connection with other MMM work; such supporting text is key for placing the presented results in context of current separation membrane studies. This point is especially important when considering the popularity of ZnO nanoparticle fillers in MMMs.

We gratefully appreciate for your valuable comment.

The appropriate changes were made in the introduction of the manuscript.

“MMMs can be filled with a variety of inorganic materials, such as metal oxides [20–25], metal organic frameworks (MOFs) [26–30], carbon molecular sieves (CMS) [31–35], carbon nanotubes (CNTs) [36–40], and zeolites [39–41]. Nevertheless, regardless of a filler type, the manufacture of MMMs is often associated with a number of difficulties, such as agglomeration of inorganic fillers, their sedimentation and insufficient dispersion [7]. The dispersion of inorganic fillers in a polymer can be improved by applying the priming method [44–46], interfacial polymerization processes [47–49], as well as additional agents to increase adhesion at the polymer/filler phase interface [50–52]. Moreover, the overall preparation of MMMs is also influenced by the size and morphology of the filler particles, the rigidity of the polymer chain, as well as the presence of affinity and interaction between the introduced phase and removed gas. Thus, the key to the successful preparation of MMMs is the selection of the suitable polymer/filler pair [53–55].”

“The effectiveness of zinc oxide as an MMMs nanofiller for CO2 recovery has already been proven in a number of works [24,56–58]. So, Farashi Z. and co-workers [24] showed that the CO2 permeability and ideal CO2/CH4 selectivity increased about 13% and 21%, respectively, at adding 10 wt. % of zinc oxide nanoparticles to the polymer matrix based on polyether-block-amide (Pebax-1657). The spherical zinc oxide nanoparticles had an average particle size of 18 nm. The membrane performance was evaluated at a pressure of 3 bars and temperature of 30 °C. In another work, Azizi N. and co-workers [58] studied effect of zinc oxide on the performance of membranes based on poly (ether-block-amide) (Pebax-1074). The authors showed that the fabricated nanocomposite membranes exhibited better separation performance compared to the neat Pebax-1074 membrane in terms of both permeability and selectivity. As an example, CO2 permeability, ideal CO2/CH4 and CO2/N2 selectivity values for the neat Pebax-1074 membrane were 110.67 Barrer, 11.09 and 50.08, respectively, while those values were 152.27 Barrer, 13.52 and 62.15 for the nanocomposite membrane containing 8 wt. % of zinc oxide nanoparticles. The spherical zinc oxide nanoparticles had an average particle size of 10-30 nm. The membrane performance was evaluated at a pressure of 3 bars and temperature of 25 °C.”

  1. Continuing with the introduction, there are several statements that are factually inaccurate or too general. As an example, the authors’ state “To implement the membrane gas separation technology, due to availability and necessary properties, as a rule, polymeric materials are used.” This statement neglects the use of carbon molecular sieves, zeolitic membrane materials, and so on.

We fully agree with you. Thank you for your comment.

Appropriate changes were made to the introduction of the manuscript.

“To implement the membrane gas separation technology, inorganic and polymeric materials are used. As a rule, inorganic membranes have exceptional chemical, mechanical and thermal stability, exhibit higher gas flows and can withstand higher pressures in contrast to polymeric ones [7]. However, their higher cost and ability to be processed into modules for large-scale applications are the main challenges in using these materials for gas separation [7,8]. In this context, polymeric membranes are the mainstay of commercialization due to their ease of processing, manufacturing, and availability.”

  1. I believe that the results and analysis reported in this work should be expanded. As it stands, several plots are presented but connection with the main motivation and novelty of the work is absent. The discussion surrounding observations should be substantially expanded.

Thank you for your valuable suggestion.

We revised the description of the results and made the appropriate changes to expand their discussion.

  1. Some of the authors’ presented discussion is speculatory. For example, “Such tendency might be due to an increase in the fractional free volume (FFV) [28] of the polymer membranes upon adding the ZnO-NPs. This fact is indirectly confirmed by the theory described in the works [25,29] and a decrease in membrane density after modification (Tab. 4).” If this statement was confirmable within statistical significance, then there would be appreciable difference in the observed permeability. Considering the logarithmic scale of Figure 5, this does not appear to be the case.

Thank you for your valuable feedback.

We do not entirely agree that our discussion is "speculatory". We assume that your opinion is related to our mistake in presenting gas transmission data on the logarithmic scale. We offer our apologies. The scale on the corresponding graphs (Figure 6 in the revised version of the manuscript) was changed.

  1. Sticking with Figure 5, there should be some discussion of the uncertainty associated with these datapoints. Without error bars on these plots, or any mention of their magnitude, it is not possible to judge if the impact of NP content and shape yields a statistically significant difference in membrane permeability.

Thanks for your constructive comment.

We fully agree with it. Error bars were added in Figure 5 (Figure 6 in the revised version of the manuscript).

  1. For table 3, I am curious about the determination of +- 1 degree for the contact angle uncertainty. Is this an estimate? A discussion about this (in the methodology) is appropriate. It is also not clear if this is 95% confidence interval or the measurement standard deviation.

The relative error for all measurements of the contact angles did not exceed 1.5%, which corresponded to an absolute error of no more than ±1°. The assessment of deviations was evaluated according to a confidence interval of at least 0.95.

  1. Figure 7 is lacking uncertainty with each datapoint. Tensile testing should be performed in replicate and results reported as an average with uncertainty. The collection of accompanying stress-strain curves should be supplied in the supplementary information. In my opinion elongation at break and tensile strength at break are somewhat strange mechanical properties to demonstrate membrane durability. I am not aware of any operational conditions where a membrane will experience 150-300% elongation. Perhaps the stress-strain curves are more appropriate for a main text figure.

Thank you for this valuable notice.

We agree that tensile strength and elongation at break are somewhat strange mechanical properties to demonstrate membrane durability. However, these parameters are generally accepted for evaluating the properties of materials based on polymers.

In the context of our work, the stress-strain curves will not carry valuable information. These curves are valuable at varying the external conditions of the measurements, such as temperature, humidity, etc. In our case, all measurements were carried out at the same parameters of the external environment.

Error bars were added in Figure 7 (Figure 8 in the revised version of the manuscript).

We appreciate for your work earnestly, and hope that the correction will meet with approval. Once again, thank you very much for your comments and suggestions.

Yours Sincerely,

Dr. Tatyana S. Sazanova.

Reviewer 2 Report

The presented article is devoted to research on expanding knowledge in the field of membrane materials science and specifically gas separation. The article is well structured, has great practical interest and can be published in the Membranes after making corrections:

Introduction

  • Since you are not using the most common modifier, it would be useful to add small paragraphs about the nature and morphology of the particles. If there are published works with the use of this particle as an additive to the polymer matrix, tell a little more about the results

Experimental section

  • Please, add some information about film forming conditions (temperature, pressure etc.)

Results and discussion

  • Have you studied how the particles are distributed throughout the volume of the membrane? Are the surfaces on different sides of the membrane vary? It would be interesting to mention this in the discussion of AFM results as well as wettability.
  • What do you think about trends in contact angles? Discuss why the hydrophilicity of Pu membranes first decrease and then increase with adding extra loadings of ZnO
  • Please, give more details about mentioned theory (P. 8 227)
  • What is the density of ZnO? In theory, the membrane density should increase with the introduction of denser particles. Please, give an explanation why the opposite is observed here.
  • Perhaps if you change the scale in Figures 5, the overall trends and maximum of permeability will be more obvious.
  • Lack of explanations for the observations made in gas separation experiments

Author Response

Dear Reviewer,

On behalf of my co-authors, we are very grateful to you for giving us an opportunity to revise our manuscript. We appreciate you very much for your constructive comments and suggestions. We have studied your comments carefully and tried our best to revise our manuscript. Thanks again to the hard work of you!

The responds to your comments are as follows:

  1. Since you are not using the most common modifier, it would be useful to add small paragraphs about the nature and morphology of the particles. If there are published works with the use of this particle as an additive to the polymer matrix, tell a little more about the results.

We gratefully appreciate for your valuable comment.

The appropriate changes were made in the introduction of the manuscript.

“The effectiveness of zinc oxide as an MMMs nanofiller for CO2 recovery has already been proven in a number of works [24,56–58]. So, Farashi Z. and co-workers [24] showed that the CO2 permeability and ideal CO2/CH4 selectivity increased about 13% and 21%, respectively, at loading 10 wt. % of zinc oxide nanoparticles into the polymer matrix based on polyether-block-amide (Pebax-1657). The spherical zinc oxide nanoparticles had an average particle size of 18 nm. The membrane performance was evaluated at a pressure of 3 bars and temperature of 30 °C. In another work, Azizi N. and co-workers [58] studied effect of zinc oxide on the performance of membranes based on poly (ether-block-amide) (Pebax-1074). The authors showed that the fabricated nanocomposite membranes exhibited better separation performance compared to the neat Pebax-1074 membrane in terms of both permeability and selectivity. As an example, CO2 permeability, ideal CO2/CH4 and CO2/N2 selectivity values for the neat Pebax-1074 membrane were 110.67 Barrer, 11.09 and 50.08, respectively, while those values were 152.27 Barrer, 13.52 and 62.15 for the nanocomposite membrane containing 8 wt. % of zinc oxide nanoparticles. The spherical zinc oxide nanoparticles had an average particle size of 10-30 nm. The membrane performance was evaluated at a pressure of 3 bars and temperature of 25 °C.”

  1. Please, add some information about film forming conditions (temperature, pressure etc.).

Thank you for your valuable comment.

Information about film forming conditions was added to the text of the manuscript.

  1. Have you studied how the particles are distributed throughout the volume of the membrane? Are the surfaces on different sides of the membrane vary? It would be interesting to mention this in the discussion of AFM results as well as wettability..

In this work, the distribution of the particles throughout the membrane volume was estimated only indirectly from the zeta potential measurements, which indicated the high stability of the obtained PU/ZnO suspensions. However, when membranes are dried, a stable suspension does not exclude the possibility of sedimentation. Therefore, we agree with you that the AFM of the reverse surface of the membranes could confirm the absence or presence of this process.

In this regard, we carried out an additional AFM study of the reverse side of the membranes. The surface roughness of the membranes on the reverse side was close to the roughness already reported in the manuscript, indicating no or minimal sedimentation.

The AFM data for the reverse side of the membranes were added to Appendix 1 of the manuscript. Appropriate explanations were included in the text of the manuscript.

  1. What do you think about trends in contact angles? Discuss why the hydrophilicity of Pu membranes first decrease and then increase with adding extra loadings of ZnO.

Thank you for your constructive suggestion.

We revised the wettability data and made the appropriate changes to expand their discussion.

  1. Please, give more details about mentioned theory (P. 8 227).

The essence of the theory mentioned in the manuscript lies in the inversely dependence between the dispersion component of the surface free energy in polymeric membranes and their FFV, which has the directly dependence with the total gas permeability. The fact is that the dispersion component is understood as the sum of van der Waals and other non-site-specific interactions between a polymer surface and an applied liquid. I.e., the dispersion component has only physical origins and depends on the morphological/supramolecular structure of the polymer. The looser the packed structure, the smaller the dispersion component of its surface energy. At the same time, the polar component of the surface free energy in polymeric membranes is understood as the sum of site-specific interactions such as electrostatic, hydrogen bonding, etc., between a polymer surface and an applied liquid. I.e., a change in the polar component can be associated with the corresponding redistribution of polar/nonpolar segments of polymer chains in membranes. Thus, an increase in the polar surface energy indicates an increase in the polar segment proportion on the polymer surface, and vice versa. In this case, the presence of polar segments of the polymer chain on the membrane surface will directly affect its interaction with separated gases. At that, plasticizing gases are more susceptible to this effect than non-plasticizing ones, since they are prone to site-specific interactions.

Appropriate explanations were added to the text of the manuscript.

  1. What is the density of ZnO? In theory, the membrane density should increase with the introduction of denser particles. Please, give an explanation why the opposite is observed here.

That’s really nice comment.

We assume that the decrease in the membrane densities with loading the ZnO NPs into the PU matrix is associated with an increase in the degree of the NPs agglomeration with an increase in their content. During the NPs agglomeration, depending on the severity of this process, the size of the inorganic phase will obviously be larger than the diameter of the initial particles. Thus, with an increase in the degree of NPs agglomeration, a forced rearrangement of PU macromolecules probably occurs, which causes a decrease in the density of the polymer composites.

Appropriate explanations were added to the text of the manuscript.

  1. Perhaps if you change the scale in Figures 5, the overall trends and maximum of permeability will be more obvious.

We gratefully appreciate for your valuable suggestion.

The scale on the corresponding graphs (Figure 6 in the revised version of the manuscript) was changed.

  1. Lack of explanations for the observations made in gas separation experiments.

Thank you for your valuable suggestion.

We revised the gas separation data and made the appropriate changes to expand their discussion.

We sincerely hope that this revised manuscript has addressed all your comments and suggestions. We appreciated for your warm work earnestly, and hope that the correction will meet with approval. Once again, thank you very much for your comments and suggestions.

Yours Sincerely,

Dr. Tatyana S. Sazanova.

Round 2

Reviewer 1 Report

I appreciate the Authors’ attention to my initial comments, which were majority satisfactory. In its current form, I do not maintain any reservations about the article being accepted for publication. I recommend the article be accepted to Membranes. I extend my congratulations on a well prepared work.